# Risk Factors for Thoracic Aortic Dissection

**DOI:** 10.3390/genes13101814

**Published:** 2022-10-07

**Authors:** Zhen Zhou, Alana C. Cecchi, Siddharth K. Prakash, Dianna M. Milewicz

**Affiliations:** Division of Medical Genetics, Department of Internal Medicine, McGovern Medical School, The University of Texas Health Science Center at Houston, 6431 Fannin Street, MSB 6.100, Houston, TX 77030, USA

**Keywords:** thoracic aortic aneurysm and dissection, acute aortic dissection, risk factor

## Abstract

Thoracic aortic aneurysms involving the root and/or the ascending aorta enlarge over time until an acute tear in the intimal layer leads to a highly fatal condition, an acute aortic dissection (AAD). These Stanford type A AADs, in which the tear occurs above the sinotubular junction, leading to the formation of a false lumen in the aortic wall that may extend to the arch and thoracoabdominal aorta. Type B AADs originate in the descending thoracic aorta just distal to the left subclavian artery. Genetic variants and various environmental conditions that disrupt the aortic wall integrity have been identified that increase the risk for thoracic aortic aneurysms and dissections (TAD). In this review, we discuss the predominant TAD-associated risk factors, focusing primarily on the non-genetic factors, and discuss the underlying mechanisms leading to TAD.

## 1. Introduction

The aorta is the largest artery in the body that originates from the heart, curves through an arch in the upper chest, and descends into the abdomen to deliver oxygen and nutrition to distal organs. A thoracic aortic aneurysm (TAA) is a balloon-like bulge or enlargement in the thoracic aorta. The natural history of these aneurysms is that they enlarge over time, until a tear forms in the intimal layer of the aortic wall typically above the sinotubular junction that leads to Stanford type A aortic dissection [1]. An acute aortic dissection (AAD) is an emergency condition that causes sudden death in up to 50% of afflicted individuals [2]. Less acutely fatal, type B AADs originate in the descending thoracic aorta just distal to the left subclavian artery. Approximately 7% of out-of-hospital sudden deaths are due to AADs, and the incidence has increased over the past 20 years for unknown reasons [3,4].

Surgical repair of root/ascending TAAs to prevent type A AAD is recommended at an aortic diameter of 5.0–5.5 cm [1]. However, the majority of patients presenting with type A AADs have diameters <5.5 cm [5,6,7,8], and type B AADs occur without significant aortic enlargement, indicating more accurate predictors of AADs are critically needed [6]. Furthermore, the incidence of AAD is 3–6 cases per 100,000 person-years in the general population, which is most likely an underestimate since up to 49% of AADs results in pre-admission death and one out of three AADs are misdiagnosed [2,9,10]. 

The predominant pathological change during aortic disease progression is aortic medial degeneration, which is characterized by elastic fiber fragmentation, smooth muscle cell (SMC) loss, and extracellular matrix accumulation of proteoglycans. Genes with rare variants that predispose to thoracic aortic aneurysms and dissections (TAD) have been extensively explored over the past 3 decades, and pathogenic variants in 11 genes have been validated to predispose to highly penetrant TAD. Identification of pathogenic variants in genes allows early diagnosis and an opportunity to initiate management to prevent AADs, including mitigating lifestyle risk factors for AADs. Additionally, up to 80% of individuals presenting with AADs do not have a known family history of aortic disease or a pathogenic variant in an established TAD gene [11], and very little is understood as to why these AADs occur. These individuals are hypothesized to harbor one or more genetic variants that predispose them to AADs, and in combination with environmental insults or a second genetic hit, trigger AADs [12]. This review will focus on the risk factors that have not been proved to cause AADs solely, and how environmental and lifestyle risk factors may combine with genetic variants to trigger AADs [13]. 

### 1.1. General Risk Factors for TAD

#### 1.1.1. Biological Sex

TAAs are more common in males than females, with a male-to-female ratio of 2–4:1 and an overall incidence of approximately 5–10 per 100,000 person-years [14]. In two of the largest population-based epidemiological studies from Sweden [15] and Canada [16], 36–39% of individuals with AADs were female. This is consistent with the 37% of women who presented with AADs in a recent International Registry of Acute Aortic Dissection (IRAD) study [17].

Sex differences in AAD have received attention due to higher in-hospital or short-term mortality in women with TAD, while both type A and B AADs show approximately twice as frequently in men [18,19,20,21,22,23,24,25]. Several studies revealed a later age of AAD onset in females compared with males, and more pharmacotherapy and aortic rupture prior to intervention in women are likely the cause of the early outcome difference [15,19,22]. Sex was not associated with short-term or long-term mortality in type B AAD patients who underwent thoracic endovascular aortic repair (TEVAR) [18,26,27]. As aortic valve-sparing root replacement and antegrade cerebral perfusion strategies during arch reconstruction increased over the past decade, in-hospital mortality due to type A AADs has decreased in both men and women [15,21,27], mitigating sex differences in short-term outcomes was found to disappear in several recent studies [20,23,27,28,29,30]. Despite these improvements, women with type A AAD are more likely to suffer neurological dysfunction due to decreased perfusion or ischemia prior to intervention and post-operation [21,22,31]. In a 2004 IRAD study, mental health outcomes were significantly worse in women, and a recent population-based study in Taiwan found that women had higher rate of depression after surgical treatment of type A AAD [27]. Men have been found to have longer intensive care unit stays, ventilator support, surgical procedure time, and cardiac-pulmonary bypass time [28,32]. Importantly, reoperation rates are significant higher in men, which has been attributed to an increased propensity for bleeding in a recent meta-analysis [28,30,32]. 

Regarding the later age of AAD presentation in women, there is growing clinical and experimental evidence that the increased risk of AAD in peripartum and postmenopausal women is caused at least in part by withdrawal of estrogen and/or progesterone [23,33]. However, the role of sex hormones in TAD is almost entirely unknown, in part due to fact that mouse studies have focused on *Fbn1^C1041G/+^* Marfan mice, which develop aneurysms but rarely dissect [34,35,36]. Estrogen is believed to mediate the major sex differences in the cardiovascular system and has shown beneficial effects in reducing hypertrophy and ischemia-reperfusion injury in mice [37,38]. Notably, a recent study found that estrogen replacement significantly attenuated angiotensin II (Ang II) plus β-aminopropionitrile (BAPN)-induced ascending aortic enlargement [39]. On the other hand, the role of progesterone is controversial. Acute administration of progesterone increases endothelial-specific nitric oxide synthase production and decreases blood pressure, which may prevent cardiovascular diseases [40,41]. However, prolonged exposure to high progesterone levels is linked to premature coronary artery disease in young males [42]. It remains unclear how or if female or male sex hormones modulate aortic wall homeostasis to affect dissection risk. 

#### 1.1.2. Pregnancy

Pregnancy leads to hemodynamic changes and rapid hormone alterations and is a risk factor for TAD. Aortic dissection in pregnancy is extremely rare, occurring in 0.2% to 1% of all female AAD cases [18,43,44], with an annual incidence of 4–12 per million maternities [43,44,45,46,47]. Kamel et al. [43] identified 36 pregnancy-related AAD cases, defined as 6 months prior to and 3 months after delivery, and 9 nonpregnancy-related AAD cases during an equivalent 9-month period exactly 1 year later, from a total of 14,999 AAD patients in 4,933,697 women with 6,566,826 pregnancies. Based on these data, pregnancy was associated with a five-fold increased risk for AAD [43]. Hypertension further increases the risk of aortic complications by up to ~three-fold during pregnancy [43]. However, the most significant risk factor for pregnancy-associated AAD are syndromic heritable thoracic aortic diseases (HTAD) such as Marfan syndrome (MFS). In these patients, the absolute increased risk for AAD is about 1000 times higher than the general population [43]. Sixty percent of women with pregnancy-associated AADs have at least one clinically identifiable risk factor for AAD, including a genetic syndrome associated with TAD (primarily MFS), a family history of TAD, or a bicuspid aortic valve (BAV) [44,48,49,50]. 

In clinical studies, the ratio of type A to type B AAD is approximately 4:1 in pregnancy [48,50,51]. The risk for these aortic events is increased up to 3 months after delivery [44,47]. Previous studies identified a significant increased risk of dissection when aortic root diameter reaches 4.5 cm in women with MFS; the risk is relatively low and women tolerate pregnancy well when the root is <4 cm [50,52,53]. Current guidelines therefore recommend vaginal delivery in women with MFS when the aortic root diameter is less than 4 cm and a cesarean delivery when the diameter is >4.5 cm, corresponding to the increased dissection risk when the maximum aortic diameter is >4.5 cm [1,54]. Importantly, the awareness of HTAD improves TAD-related prognosis of pregnant women, whereas that lack of a diagnosis of HTAD prior to aortic event occurrence is likely associated with early delivery, surgical intervention, and type A AAD [44,51,55]. In a prospective study [49], 189 out of 5739 women were enrolled with pre-existing TAD prior to pregnancy, including 100 (53%) MFS, 49 (26%) cases with BAV, 16 (8%) Turner Syndrome, 4 (2%) vascular Ehlers-Danlos syndrome cases, and 20 (11%) TAD cases without prespecified HTAD or associated congenital heart defect. Only four MFS patients presented AADs (three type A and one type B AAD), and there was no maternal or fetal mortality. In contrast, the latest update from the IRAD study identified a total of 29 pregnancy-related aortic events and 20 (69%) cases had an underlying HTAD; however, about half of the patients (7 in 15 reported) were not aware of their HTAD diagnosis until after dissections occurred [44]. Taken together, these findings validate the efficacy of current guidelines and highlight the importance of early recognition, diagnosis, and preconception counselling in patients with risk factors for TAD to prevent AADs and achieve better pregnancy outcomes. 

As to why pregnancy increases aortic complications, there is a 30–40% increase of cardiac output during pregnancy [56], leading to increases in both heart rate and stroke volume. The increase of heart rate occurs by the end of first month and plateaus at an increase of 10–15 beats per minute by 28–32 weeks of pregnancy. Stroke volume increases by 6 weeks and progressively increases through the second trimester. Cardiac output further increases up to 50% and 80% above pre-labor level during labor and immediate postpartum period, respectively. All these changes decrease to pre-labor values 24–72 h and return to nonpregnant levels within 6–8 weeks postpartum [57]. Further detailed timetables of AAD onset during pregnancy are required to study if the prevalence of pregnancy-associated aortic events is consistent with these hemodynamic alterations during these time courses. β-blockers decrease heartbeat rate and force generation; thus, they are widely accepted to treat pregnant patients with HTAD. Immer et al. suggested that β-blockers should be administrated to all pregnant patients with aortic root diameter >4 cm with continued use up to 3 months postpartum [50]. Importantly, a prospective study found no maternal or fetal mortality or differences in birth defects, and a similar average fetal birth weight in pregnancies in which the mother was treated with or without β-blocker [49], suggesting that β-blockers are safe for use during pregnancy. 

#### 1.1.3. Circannual and Circadian Factors

Similar to other adverse cardiovascular events, such as acute myocardial infarction, acute pulmonary thromboembolism, and cerebrovascular accidents, AAD has chronobiological patterns [58,59,60]. In 1997, Gallerani et al. [61] analyzed hour of symptom onset from 67 cases and identified that AADs has a circadian pattern, with a primary peak of onset at around 10 a.m. and a secondary peak at around 8 p.m. In 1999, Manfredini et al. [62] studied monthly data from 85 AAD patients and demonstrated a circannual pattern with a peak of maximal occurrence in January/winter/cold season. Since then, increasing numbers of large-cohort studies confirmed these findings of AAD onset patterns from different locations of the northern hemisphere [58,59,60,63,64,65]. Despite the lack of AAD data from the southern hemisphere, two recent studies revealed similar seasonal pattern in cervical artery dissection in Australia and the latest study showed consistent increased risk of cervical artery dissection in cooler months [66,67]. These findings together indicate circannual and circadian causes are important in the pathogenesis of AAD. Studies did not show any differences of in-hospital adverse clinical events or mortality with different seasons or different time-of-day onset. 

A few early studies identified a relative higher AAD risk on Monday than other days of a week [59,60,68], but this pattern was not confirmed in more recent studies [65,69,70]. However, there is a significantly higher in-hospital mortality rate in patients admitted for AAD or rupture on weekends compared with weekdays [70,71]. In fact, similar patterns have been observed for myocardial infarction, pulmonary embolism, acute heart failure, and stroke [72,73,74,75,76,77,78]. Takagi et al. [71] analyzed 11 studies, including a total of 166,195 patients, and concluded that weekend admission/surgery for ruptured aortic aneurysm and AAD may be associated with increased mortality, mainly due to the “off-hour effect” characterized as a shortage of staff and lack of experience clinicians, inadequate subspecialty care and limited therapeutic or diagnostic procedures. 

Kobza et al. [60] observed that type A AADs peak in winter and type B AADs peak in spring. Recent studies found that admission rates and in-hospital mortality due to type A AADs increase during influenza season, while type B AAD rates are not associated with influenza [79,80]. Animal studies have shown that influenza virus mRNA could be detected in aortic tissues after exposure mice to H3N2 virus, together with increased mRNA levels of *Il1*, *Il6*, and *Mcp1*, which are predominant inflammatory drivers of aortic remodeling [81]. Further studies are required to validate the correlations between TAD prevalence and seasonal influenza, and whether flu vaccination will impact the incidences. 

The underlying causes and mechanisms of circannual and circadian risks for TAD are complex and largely unknown. Increased sympathetic nerve activity is believed to be the major mediator in the occurrence of TAD chronobiological periodicity [82], because: (1) sympathetic rhythmicity is strongly associated with blood pressure and heart rate regulation [83], especially before and after awakening [84]; (2) cold exposure is a major trigger for sympathetic activation [85]. Sympathetic activation causes vasoconstriction of both the peripheral and visceral arteries, leading to high peripheral resistance and increase of blood pressure and heart rate, which subsequently altering the hematologic and hemodynamic properties [86] and shear stress on the intima, favoring aortic injury. 

#### 1.1.4. Genetic Variants 

Approximately 20% of families with TAD exhibit an autosomal dominant inheritance pattern, indicating Mendelian inheritance of a pathogenic variant conferring a highly penetrant risk for TAD [11]. In 2018, 11 out of 53 candidate genes were designated as causal for HTAD [13]. Cascade screening of families who are found to have mutations in any of these 11 genes is clinically indicated to prevent premature deaths of affected relatives due to TAD. The genetic risks for TAD extend from highly rare penetrant variants that trigger disease in almost all individuals carrying the alteration to common and lower penetrant variants more commonly found in the general population that confer a lower risk for disease, which has been extensively discussed in previous reviews [11,87,88,89] and will not be presented in this paper.

#### 1.1.5. Bicuspid Aortic Valve (BAV)

BAV is the most common adult congenital heart malformation with an overall prevalence of 1% but is three times more frequent in males (1.5%) than in females (0.5%) [90]. The prevalence of BAV is also significantly higher in European populations than in populations of African ancestry [91]. BAV is primarily inherited as an autosomal dominant trait with incomplete penetrance and variable expressivity [92]. BAV inheritance is best explained by a complex genetic architecture involving many different interacting genes [93].

Due to the common embryologic origin of the aortic valve, left ventricular outflow tract and proximal aorta, BAV frequently co-exists with other left-sided congenital heart lesions such as coarctation, mitral valve abnormalities, ventricular septal defects, and most frequently TAAs in more than one-third of patients [94]. BAV-associated aneurysms are predominately located in the ascending aorta but can also involve the aortic root. Aneurysmal dilation of the aortic root in BAV patients is associated with an earlier clinical presentation and more rapid progression of aortic disease, particularly if other congenital heart lesions or extracardiac abnormalities are present [95]. Some BAV cases with these features may have chromosomal lesions (Velocardiofacial syndrome or Turner syndrome) or mutations in causal HTAD genes [96]. Shear stress on the ascending aortic wall as measured by 4D flow cardiac magnetic resonance is a promising metric to predict the rate of aortic dilation in BAV patients [97]. 

BAV is enriched 5–10-fold in rare syndromic forms of HTAD such as Loeys–Dietz syndrome that do confer a high risk for AAD [98]. This may explain why the prevalence of BAV is increased in AAD cohorts compared to the general population (3.2% in IRAD [99]). However, more than 90% of BAV cases occur as isolated non-syndromic lesions, and in this group, the overall risk of AAD appears to be very low (0.4% over 15 years), even with significant aortic dilation [100,101]. Current clinical guidelines reflect these observations by setting surgical thresholds for aneurysms in non-syndromic BAV cases as high as 5.5–6.0 cm [102,103]. 

#### 1.1.6. Geography and Ancestry

The incidence of aortic dissection ranges from 3 to 6 cases per 100,000 person-years globally [2,10]; however, data from large-scale registries have demonstrated that the frequency of dissections reported, demographics, management and outcomes, are variable depending on the ancestral populations studied and geographic location of data collection.

With the largest population in the world, China launched Registry of Aortic Dissection in China (Sino-RAD) project in 2011 and published a comparison study with previously published data from IRAD in 2014 [104]. Compared with IRAD which primarily includes patients from North American and European countries, Sino-RAD data showed an early age of dissection onset (51.8 ± 11.4 vs. 63.1 ± 14.0, *p* < 0.01), higher male proportion (77.8% vs. 65.3%, *p* < 0.01), and a lower incidence of hypertension (58.7% vs. 72.1%, *p* < 0.01) [104]. Another notable difference between the IRAD and Sino-RAD data was a higher frequency of type A AADs, accounting for 66.7% of the total AADs reported, in the 20-year IRAD update [105] compared with 42.9% in China [104]. In contrast, demographic characteristics, clinical presentation, and outcomes reported from a single center in Australia were consistent with IRAD data [106]. Overall, the age of type A AAD onset is consistently earlier in patients from East Asian countries [104,107,108,109] compared with European, North American and Australian registries [105,106,110,111]. The male-to-female sex ratio is higher in most dissection registries, ranging from 2:1 to 3:1, with the exception of data from Japan and South Korea showing a 1:1 sex ratio (Table 1). Finally, there is a lower prevalence of concomitant hypertension (51–53%) reported in studies from China, South Korea, German Registry for Acute Aortic Dissection Type A (GERAADA), and the Nordic Consortium for Acute Type A Aortic Dissection (NORCAAD), while hypertension occurs in 72–75% of patients with type A AAD in studies from IRAD, Japan, Taiwan, and Australia (Table 1). 

Racial disparities and inequities in health, clinical outcomes and medical practice in patients with aortic dissection have received more attention in recent years. In particular, notable differences in clinical presentation, management and health outcomes have been observed in patients who self-identified as Black compared with White in the USA. An IRAD study that included 189 self-reported Black patients and 1165 White patients from 13 USA centers and found an association between Black race and earlier age of dissection onset, more type B AADs, and increased rates of hypertension and diabetes [112]. These characteristics have been repeatedly reported in other studies [113,114]. A study in the USA looked at 60,784 patients in the Nationwide Inpatient Sample database who had TEVAR for all types of thoracic aortic diseases, and found that Black, Hispanic, and Native American patients were more likely to receive TEVAR compared with White patients [115]. Other studies demonstrated that Black race was independently associated with a lower risk of 30-day mortality [116], and this survival advantage extended from 1 to 5 years after TEVAR for type B AAD [117]. However, reintervention rates after TEVAR were higher among Black patients compared with White patients [117]. Importantly, compared with Whites and Blacks, Hispanic race was independently associated with increased in-hospital mortality (RR 2.57; 95% CI, 1.25–5.25; *p* = 0.01) after open repair of unruptured thoracoabdominal aneurysms [114]. 

Overall, multicenter registries incorporating data from patients in different geographic regions with diverse ancestries has enabled research into associations between TAD incidence and other demographic and clinical characteristics, and geographic location, ancestral origin and self-reported race of patients with TAD. However, the collection and reporting of race, ethnicity and ancestry data are not consistent across studies, and interpretation of their role in dissection risk and clinical outcomes requires further investigation of genetic factors, dietary habits, stress burden, drug use, and other social determinants of health to delineate new risk factors and improve TAD management.

### 1.2. Modifiable Risk Factors for AADs

#### 1.2.1. Hypertension

Hypertension is the most common comorbidity condition in patients with AADs with prevalence from 45% to 100% in previous observational studies [2,17,118,119]. In a recent autopsy study, Huynh et al. [120] found that about 84% cases died of aortic dissection showed left ventricular hypertrophy, a marker of preexisting hypertension. The first prospective population-based study identified 67.3% of AAD patients diagnosed with hypertension. In this study, the prevalence of hypertension increased from 42.9% in the first 3 years to 85.7% in the last 3 years of the 10-year study due to increased blood pressure screening in primary care, suggesting a high risk of poorly controlled hypertension prior to AAD events at the beginning of the study [2,121]. Importantly, the study showed that only 67.3% of the patients were on antihypertensive medication during the 5 years prior to dissection, and the proportion analyses of all the blood pressure data showed that 61.9% of subsequent blood pressure readings were >140/90 mmHg despite the majority of patients being treated with combined antihypertensive therapy [2]. Moreover, during the 5 years before aortic events, premorbid systolic blood pressure was significantly higher in patients with type A AAD that were immediately fatal than in those who survived to admission (151.2 ± 19.3 vs. 137.9 ± 17.9 mmHg; *p* < 0.001) [2]. These data indicate that uncontrolled hypertension, rather than hypertension itself, is correlated with AAD incidence, and better control of blood pressure would likely reduce both AAD incidence and associated mortality. Similarly, Landenhed et al. [122] found that hypertension was present in 86% of individuals who subsequently developed aortic dissection during up to 20 years follow-up for aortic diseases. Importantly, the study estimated the population-attributable risk of hypertension for aortic dissection was 54%, and the dissection risk of individuals with hypertension was about four times higher than normotensive people in general population (21 vs. 5 per 100,000/year). Similarly, the study provides further evidence supporting previous prospective study that about 50% of aortic dissection events could be prevented by effective antihypertensive treatment [2,122]. Surprisingly, in patients with moderately dilated ascending aorta (4.5–5.5 cm), hypertension was not statistically associated with aortic dissection or rupture (*p* = 0.086) [123]. However, high blood pressure has been redefined in 2017 and 130/80 mmHg is the new criteria for hypertension diagnosis [124], and future studies will further evaluate the risk of hypertension in AAD. 

#### 1.2.2. Dyslipidemia

Dyslipidemia has been well documented to advance disease progression of abdominal aortic aneurysm (AAA) and rupture, and statins treatment could significantly reduce rupture risk in AAA. Studies have revealed that dyslipidemia is also a common comorbidity in patients with TAD [119,122,125], however, only two recent studies found that it is associated with aortic enlargement and AADs. In a prospective 20-year follow-up study, Landenhed et al. [122] found that lower apolipoprotein A1 levels were associated with aortic dissection. In another prospective study, Yiu et al. [125] found that hyperlipidemia (*p* = 0.0321) was positively correlated with the growth rate of aortic arch aneurysm. Hypercholesterolemia promotes atherosclerosis of the aorta, which is initiated with oxidative LDL-overloaded macrophages in the intima and buildup by continuous fatty deposition. There is increasing evidence showing that SMC phenotypic modulation driven by cholesterol is a major contributor to atherosclerosis [126,127,128,129,130]. This modulation of SMCs may also contribute to thoracic aortic disease, and anti-hyperlipidemic therapy may have a protective effect against TAD and rupture, in addition to abdominal aortic disease. In a 26-year retrospective study, Stein et al. recruited 1560 TAA patients with (24%) or without (76%) statins treatment and found that statin therapy reduced the yearly rate of dissection, rupture and death in patients with ascending and descending TAA (*p* = 0.001–0.01), increased the interval from diagnosis to aortic events onset or surgery, and reduced the total surgical treatment rate by 10% compared with non-statin treatment TAA patients [131]. Importantly, pravastatin showed equal beneficial effect as losartan in attenuating aortic root dilation in the Marfan mice and further histology and ultrastructural analyses showed preserved aortic medial elastin volume and decreased SMC rough endoplasmic reticulum activity in statin-treated aorta, respectively [132]. These findings suggest that dyslipidemia may be a significant risk factor for TAD, and the inhibition of cholesterol synthesis with statins may be beneficial in preventing AAD incidence in TAD patients, but further prospective randomized controlled trials are needed. 

#### 1.2.3. Aortitis

Aortitis refers to inflammation of the aorta and can be divided into infectious and non-infectious categories. Infectious aortitis is caused by specific organisms, either from intraluminal endocarditic vegetations or external adjacent infectious process, most commonly *Salmonella*, *Staphylococcus*, *Strptococcus*, *Treponema pallidum*, fungal, and mycobacterial [133,134,135,136]. Because diagnosing infected tissue is rarely possible at TAD onset, uncontrolled sepsis causes about 21–44% in-hospital mortality in patients with infectious aortitis despite of combined antimicrobial and surgical interventions [137]. In contrast, syphilitic aortitis usually occurs decades after primary infection, and the aorta enlarges due to inflammatory responses and fibrosis and leads to aortic aneurysm and rupture but rarely dissection [138,139,140]. In two recent case reports, ascending aortas were involved by the syphilitic process in all patients [139,140], indicating a strong association between syphilis infection and TAD. 

The most common non-infectious causes are primary large vessel vasculitis (LVV), giant cell arteritis (GCA) and Takayasu arteritis (TAK) [135,141]. Women are more commonly diagnosed with both GCA (3:1) [142,143] and TAK (6:1) [144,145], and often present with systemic manifestations, i.e., fever, anorexia, weight loss, night sweats, arthralgia/myalgia, and/or increased inflammatory indexes. The inflammatory response within the vascular wall can cause both occlusive and aneurysmal arterial diseases, and distal organ malperfusion [146]. GCA is typically diagnosed in patients ≥50 years of age and most commonly affects the medium and large vessels, with involvement of the aorta and its primary branches in ~50% of patients [147,148]. On the other hand, TAK tends to occur in patients younger than 40 years (median age of onset ~35 years) and affects one or more aortic segments, often in combination with branches such as carotid, subclavian or renal arteries [145]. Histologically, both GCA and TAK are granulomatous diseases affecting all layers of the aortic wall and characterized by infiltrated lymphocytes, macrophages, and multinucleated giant cells [148]. TAK is more commonly associated with intimal and adventitial thickening with intraluminal stenosis, leading to ischemic diseases, previously termed “pulseless disease”. In GCA, the aorta has medial elastic lamina destruction and necrosis, which is associated with increased risk of aortic enlargement and rupture [149]. Distinct from GCA and TAK, aortitis may occur in a single aortic segment without clinical or histopathological evidence of systemic disease, which is termed clinically isolated aortitis (CIA) [148,150]. Current limited demographic data suggest that the thoracic aorta is most affected, especially the aortic arch and descending thoracic aorta. 

Despite significant heterogeneity in aortitis definition and manifestation over-lapping, several studies attempted to investigate the aortic complication rates between different aortitis subtypes. Miller et al. [149] found similar ascending aortic complication rates between CIA and LVV patients in an 83-month follow-up study. Similarly, Clifford et al. [151] reported outcomes from 196 patients (66% classified as CIA) with histologically proven aortitis following surgical repair, 44% CIA and 40% LVV patients developed new vascular lesions and required re-intervention. Importantly, the availability of high-resolution vascular imaging has improved the identification of aortitis. Fluorine-18-fluorodeoxyglucose positron emission tomography/computed tomography (18F-FDG PET/CT) has a role in diagnosing aortitis, showing an increase in FGD uptake by inflammatory cells in the aorta [152,153,154]. Ferfar et al. [155] followed up 353 patients with aortitis for 52 months and found that AADs occurred in 5%, 2%, and 10% of patients with GCA, TAK, and CIA, respectively. These data indicate that patients with aortitis are at risk for future aortic complications, particularly in those with CIA. Additionally, other non-infectious diseases may also increase the risk of aortic diseases, including ankylosing spondylitis, Behçet’s disease, Cogan’s syndrome, granulomatosis with polyangitis, IgG4-related disease, relapsing polychondritis, rheumatoid arthritis, sarcoidosis, and systemic lupus erythematosus [135]. The pathogenesis and triggers of AADs in these aortitis subtypes remain largely unknown and require further studies. 

#### 1.2.4. Obstructive Sleep Apnea (OSA)

OSA is the most common sleep disorder, which provokes the remodeling of respiratory and cardiovascular systems due to intermittent hypoxia caused by frequent obstruction of the upper respiratory tract during sleep. Symptoms of OSA include snoring, apneic attack, and daytime hypersomnia. Accordingly, the frequency of apneas, hypopneas, oxygen desaturation, and intermittent hypoxia and re-oxygenation are sensitive indicators for diagnosing and assessing the severity of OSA. The prevalence of OSA has been estimated at 22% in men and 17% in women, and the prevalence has increased over the past two decades, partially due to increased rates of obesity [156,157]. Importantly, cohort studies including the Sleep Heart Health Study have consistently demonstrated that over 50% of individuals with OSA have hypertension [158,159,160], a major risk factor for cardiovascular disorders, including AADs. Since Sampol et al. [161] observed a significant correlation between OSA and TAD in 2003, evidence from retrospective studies and meta-analysis support OSA as an independent risk factor for aortic root dilation, type A and B AAD [162,163,164], in the general population and in patients with genetic disorders, including MFS and Loeys–Dietz syndrome [165,166,167,168]. A prospective study followed up 44 MFS patients for 24–36 months, five of fifteen patients with OSA had aortic root replacement or aortic rupture while none of the 29 patients without OSA had aortic event within the follow-up period, further analysis showed a significantly decreased event-free survival in patients with OSA compared to those without OSA (*p* = 0.012) [166]. Gaisl et al. [169,170] reviewed two prospective studies and found that the prevalence of OSA was significantly higher in patients with TAA (apnea-hypopnea index ≥ 5/h) compared to a 2-to-1 matched control population (63% vs. 47%; odds ratio 1.87, 95% CI 1.05–3.34; *p* = 0.03), and the odds ratio was 3.25 (95% CI 1.65–6.42; *p* < 0.001) in patients with moderate and severe OSA (apnea-hypopnea index ≥ 15/h). After follow up the TAA cohort for an average of 3 years, Gaisl et al. [169] found a strong positive association of aortic root and ascending aortic enlargement with OSA. However, further prospective, randomized match-control trials are required to evaluate the causal relationship between OSA and TAD. 

Additionally, OSA is a treatable respiratory disorder through the use of continuous positive airway pressure (CPAP) therapy. Several case reports have shown beneficial effects of CPAP in TAD patients in slowing the aortic expansion [167,171,172]. However, there has been no evidence that CPAP therapy for OSA can reduce the incidence of aortic diseases. 

OSA-induced pressure differences, including increased intra-aortic blood pressure and decreased negative intrathoracic air pressure, have been widely considered mechanical stressors on the aorta. However, no significant difference was found when comparing the location of entry in patients with OSA between thoracic and abdominal aortic dissection subgroups [173], suggesting that mechanisms other than negative intrathoracic pressure involved in the pathogenesis of TAD. It is believed that intermittent hypoxia and re-oxygenation is the potential underline contributor of systemic vascular remodeling caused by OSA, leading to sympathetic nervous system activation and subsequent hypertension or increased systemic oxidative stress [174]. Interestingly, intermittent hypoxia and re-oxygenation induction significantly augmented aortic rupture death in BAPN plus Ang II-induced aortic dissection mouse model [175]. Hif-1α, a major transcriptional factor in response to hypoxia, was significantly upregulated and activated in the triple-treatment group when compared with BAPN alone or BAPN + Ang II group, which was consistent with exaggerated reactive oxygen species production both in the aortic tissue and explanted SMCs [175]. Additionally, administration of a specific HIF-1α inhibitor, KC7F2, delayed the aortic event and significantly decreased the mortality due to aortic rupture [175]. 

Taken together, OSA has been demonstrated as an independent risk factor for TAD in the general population, and CPAP should be used to treat OSA, even in the absence of a controlled clinical trial.

#### 1.2.5. Fluoroquinolone

Fluoroquinolone (FQ) is one of the most commonly prescribed antibiotic classes mainly due to the broad spectrum and tolerance for long-term administration [176]. The common side effects of FQ are nausea, vomiting, peripheral neuropathy, dysglycemia, and arrhythmias, with more serious collagen-disruption associated complications including tendon rupture and retinal detachment [177,178,179]. In 2015, two independent observational studies [180,181] found that FQ was associated with an approximately 2- to 3-fold increased risk of TAD, which raised safety concerns about FQ administration. Additionally, two propensity score-matched studies provided further evidence that use of FQ within 60 days was associated with high risk of aortopathy [182,183]. 

In these early studies, patients who received FQ were compared with those who did not receive FQ or with those who receive amoxicillin targeting different types of infection from FQ. While only patients with FQ-indicated infections were studied, Dong et al. [184] found that FQ was not associated with an increased aortic diseases risk when compared with other antibiotics, and aneurysms and AADs were more often seen in patients with FQ-indicated sepsis and intra-abdominal infections, suggesting the underlying infection rather than the substances was the trigger for TAD. A side-by-side publication [185] found similar increased risk for aortic aneurysm and dissection in patients receiving FQ for pneumonia compared to those receiving azithromycin (0.03% vs. 0.01%); however, FQ-associated risk for aortic disease was absent in patients with urinary tract infection. Importantly, when patients who had a diagnosis of aortic aneurysm or dissection were excluded by baseline imaging from the urinary tract infection cohort, no increased rate of aortic disease was observed in patients treated with FQ compared to amoxicillin [185], indicating a potential surveillance bias in previous studies. Consistently, without aortic disease history, short-term ciprofloxacin prophylaxis after prostate biopsy did not increase the overall risk of aortic aneurysm or rupture, however, a significantly increased risk of aortic aneurysm was noted in patients with high-risk prostate cancer most likely due to detection bias by more frequent imaging in these patients [186,187]. These studies raised critical concerns about confounding or surveillance bias in previous studies, and consequently, whether FQ triggers de novo aortopathy remains unclear, leaving the uncertainty to physicians for prescribing FQ to patients [188]. 

To answer the question whether FQs are associated with aortic aneurysm and dissection in patients with specific infections or pre-existing aortic defect, Chen et al. [189] performed a retrospective study on FQ or amoxicillin exposure to patients with existing aortic disease and found that FQ was associated with a higher risk of aortic-related events and death. Consistent with this clinical finding, a preclinical study using high-fat diet treatment plus Ang II infusion-induced TAD mouse model found that ciprofloxacin exposure significantly increased the aortic incidences and augmented aortic wall disruption [190]. In advance to the two studies [184,185] raising concern about confounding in previous findings, Newton et al. [191] looked at all FQ-related indications in a population-based study and found that FQ was associated with about 20% increase of risk of AAA and iliac artery aneurysm, and all adults were more likely to receive intervention. Further stratified analyses found that adults of 35 years or older were at increased risk of arterial aneurysm formation [191]. Since only outpatients were studied, whether in-hospital patients with more severe infections will show a higher risk of aortic disease remains to be assessed [191]. In addition to these adult-only studies, Yu et al. [192] evaluated the safety of using FQ in 0–18 years old population in Taiwan and found that FQ exposure did not increase risk of collagen-associated adverse effects, including tendons rupture, retinal detachments, gastrointestinal tract perforation, aortic aneurysm or dissection. 

Noting that patients with genetic variants were either completely excluded or presenting limited proportion in previous studies. A recently preclinical study found that ciprofloxacin accelerates aortic enlargement and promotes dissection and rupture in the *Fbn1^C1041G/+^* mouse model [193]. Taken together, in 2018 and 2019, the United States Food and Drug Administration and the European Medicines Agency requested that manufactures update FQ safety information related to the increased risk for TAD and its complications, and warned that the use of these drugs in patients at risk for aortic complications in clinic, including MFS and other related conditions, respectively [194].

Finally, several in vivo and in vitro studies have consistently found that FQ exposure regulated extracellular matrix remodeling-related enzyme expression or activity, including increasing matrix metalloproteinases (MMPs), decreasing tissue inhibitors of MMPs and lysyl oxidase [190,195,196]. However, knowledge gaps remain exist between these observations and the direct-action mechanism of FQ, which mainly targets two essential bacterial type II topoisomerase enzymes, DNA gyrase and DNA topoisomerase IV [197]. Interestingly, early studies found imbalanced cellular calcium homeostasis was involved in pharmacological effects of FQ [198,199]. Noting that calcium-channel blockers are effective antihypertensive drugs and are frequently administrated as an alternative to β-blockers; however, the use of calcium-channel blockers was associated with increased 30-day mortality after acute or elective abdominal or thoracoabdominal aortic aneurysm surgery [200]. When compared to other antihypertensive agents, Doyle et al. [201] found that calcium-channel blockers significantly increased the risk of AAD incidence in MFS patients, and the need for aortic surgery in patients with various forms of HTAD beyond MFS. It remains unclear if intracellular calcium imbalance-mediated cellular energy metabolism and calcium homeostasis are involved in FQ-associated disruption of aortic integrity. 

#### 1.2.6. Cocaine

Insufflation of cocaine powder is the most widely used method due to its longer duration time after each use, and cocaine users have shown about six-fold higher risk of all-reason mortality when compared to the general population [202]. Since 1987 [203], several case reports and retrospective studies have linked cocaine with AAD [204,205]. 

Hsue et al. [206] first reported 37% (14/38) of patients with AAD had history of cocaine use within 24 h of symptoms onset. This study reported an increased prevalence of cocaine-related AADs among young (41 ± 8.8 years), uncontrolled hypertensive (11/14, 79%), and smokers (14/14, 100%). Despite the fact that this is an inner-city location and single-center based study, which showed the highest prevalence of cocaine use in all the studies so far, six following studies confirmed these observations with a wide variation of cocaine use, ranging from 0.5% to 28%, and five of these studies reported that the majority of the cocaine users were male (71–88%) [207,208,209,210,211,212]. A previous IRAD report showed only 0.5% (5/921) patients with aortic dissection had a history of cocaine use in 2002 [207], while this proportion increased to 1.8% (63/3584) in 2014 [210]. Noting that IRAD study only recruited in-hospital patients while numbers of case report and autopsy study proved that a large number of cocaine users died of aortic dissection and rupture outside hospitals [204,213,214,215,216], leading to the concern about underestimated cocaine-use prevalence in both IRAD studies. However, the updated IRAD study found that cocaine-use patients had a higher risk of type B AAD (2.4%, 30/1252) than type A AAD (1.4%, 33/2332), which is consistent with a previous study [209]. Furthermore, Yammine et al. [212] found that cocaine users were more likely to present with larger falser lumen diameters and higher risk of endoleaks among all the type B aortic dissection patients. This study also found more reinterventions in those cocaine-use patients, which confirmed the updated IRAD finding that a higher rehospitalization rate among the cocaine-use patients which possibly related to continuing cocaine use [210]. However, the long-term outcome between cocaine-use and noncocaine-use AAD patients are controversial mainly due to small cohort and loss of follow up in previous studies [209,210], which requires further large-scale study to reveal. Taken together, the current studies consistently found that cocaine use is associated with increased AAD risk in young hypertensive males who smoke. 

The common underline mechanism of cocaine-associated TAD and other cardiovascular diseases, such as coronary artery infarction, hemorrhagic or ischemic stroke, and atherosclerosis is cocaine-mediated sympathomimetic effect. Cocaine activates central sympathetic outflow directly [217] and blocks reuptake and breakdown of norepinephrine and dopamine at the presynaptic clefts, leading to constitutive activation of postsynaptic receptors in the peripheral sympathetic nerve [218,219]. Additionally, peripheral sympathetic nerve activation may stimulate adrenal release of catecholamines, which further activate the sympathetic nerve systemically via an endocrine effect. The excessive activated sympathetic nerve system and release of catecholamines mainly act on cardiomyocyte and SMC mainly via the β and α adrenergic receptors, respectively [217]. Stimulation of these receptors will cause the following effects related to TAD: (1) A rapid elevation of blood pressure; (2) increases of heart rate and stroke volume, which accelerate the shear stress over the aortic wall and cause intimal injury; and (3) SMC hypercontraction leads to vasospasm which may initiate thrombus formation in the microcirculation system, such as the vaso vasorum supplying blood to the aorta [220]. Importantly, increased oxidative stress and mitochondria dysfunction may trigger cardiotoxicity, and a similar response may occur in the hypercontractile SMCs [221,222]. Additionally, cocaine-induced endothelial injury characterized as decreased vasodilator nitric oxide in ex vivo tissues, and increased vasoconstrictor endothelin-1, prothrombotic factor, and even circulating endothelial cells in cocaine-use patients [223,224], suggesting a significant role of endothelial dysfunction in cocaine-caused cardiovascular incidents. 

#### 1.2.7. Other Acquired Conditions

Several acquired factors have been reported to be associated with TAD, including: (1) intrinsic factor, weightlifting [225,226,227]; (2) extrinsic factors, (i) trauma [228]; (ii) iatrogenesis [229,230]; and (iii) misdiagnosis [9]. The latter two extrinsic factors are relevant to medical knowledge and skill trainings. Nevertheless, the awareness of these acquired TAD risk factors will be beneficial for disease management as well as lowering the incidence rate. 

#### 1.2.8. Protective Factor—Diabetes

Diabetes is a well-established risk factor for coronary and cerebrovascular diseases. Surprisingly, the prevalence of diabetes with AAA ranged from 6 to 14% which is significantly lower than 17–36% in those without AAA [231], and diabetes was found independently associated with reduced AAA growth in a 3-year follow-up study [232]. In 2012, Prakash et al. first found an inverse association between diabetes and TAD hospitalization rate [233]. Similarly, Takagi et al. performed a meta-analysis which recruited 11 studies with a total of 47,827 patients with TAD and found that diabetes presented in 2.3–15.8% of TAD patients while the range was 8.9–35.9% in control cohorts (OR 0.43; 95% CI, 0.31–0.59; *p* < 0.00001) [234]. 

Chen et al. found that the aortic root dilation was less prevalent (2.75% vs. 9.63%; *p* = 0.025) and the mean aortic root dimensions were significantly smaller in diabetic patients than in nondiabetic patients [235]. In a retrospective study, Theivacumar et al. recruited 2062 TAA and AAA patients and found that diabetes prevalence in ruptured and non-ruptured aneurysm were 5.6% and 13.2%, respectively (OR 0.42; 95% CI, 0.23–0.75; *p* = 0.004). Additionally, Tsai et al. found that the risk of TAA rupture in patients with advanced diabetes (ICD-9-CM diagnosis codes 250.4–250.9) decreased for nearly 50% compared to nondiabetic cohort in a population-based study [236]. These findings provide evidence that diabetes acts as a protective factor during TAD progression and rupture incidence. 

It is worth noting that the prevalence of diabetes does not differ between patients with type A and type B AADs [237]. Despite a low prevalence of diabetes in patients with TAD, elective surgical repair of proximal aorta for diabetic TAD patients should be paid further attention because the in-hospital death or permanent stroke rates are >three times higher than nondiabetic TAD patients [238]. On the contrary, type B aortic dissection patients received TEVAR showed significant lower mortality and complications in the diabetic group than in the normal glucose group in a 3-year follow-up study [239]. 

The underlying mechanism of hyperglycemia-associated beneficial effects in TAD is not fully understood. Given the fact that *ACTA2* variants predispose to both TAD and early onset stroke and coronary artery disease [240], suggesting that SMCs are potentially the predominant cell type that involved in both large elastic artery enlargement and small muscular artery occlusion; however, it is unclear but interesting if a common signaling pathway is involved. In vitro study has shown significantly increased capacities of proliferation, adhesion, and migration in SMCs from patients with diabetes compared to nondiabetic SMCs, which is believed to promote plaque formation and the major cause of small artery occlusion [241]. Interestingly, our unpublished data show that SMC-specific *Pdgfrb* deficiency augments aortic root enlargement in mice with *Acta2*^−/−^ background. These results indicate a possible protective role of SMC proliferation in inhibiting aortic dilation under hyperglycemia condition. Moreover, hyperglycemia enhances collagen glycation and networking which counteracts proteolysis driven by MMPs, while glycated type I collagen further decreases MMPs and interleukin-6 generation in activated monocytes [232] which are another major type of cells that contribute to extracellular matrix degradation and remodeling involving in both atherosclerotic plaque and TAD progression. 

### 1.3. Aortic Dimension

#### 1.3.1. Dilatation (Circumferential Enlargement)

Increasing aortic diameter is a demonstrated risk factor for TAD, easy assessment of this parameter by various imaging methods concludes a diameter of 5.0–5.5 cm as an inflection point for elective surgical repair based on predictive risk analyses of dissection, rupture, and death [1]. However, accumulating evidence shows that many patients with acute TAD have diameters <5.5 cm [6], indicating an urgent need of lowering this prophylactic indicator for surgical treatment of TAD patients without connective tissue disorders or aortitis. 

Parish et al. [7] found that 62% type A AADs occurred at ascending aortic diameters <5.5 cm and 42% at diameters <5.0 cm, which are extremely similar to the proportions found from IRAD: 59% and 40% of dissections occurred at ascending aortic diameters <5.5 cm and <5.0 cm, respectively [6]. Regarding a rapid expansion of aortic diameter after dissection onset and a non-natural condition in previous studies, Kim et al. [123] performed competing risk analyses on 4654 non-syndromic adults with maximal baseline ascending aortic diameters of 4.0 to 5.5 cm and demonstrated a linearized incidence of aortic dissection and/or rupture of 0.1% per patient-year, and initial aortic diameter was an independent predictor of aortic event (HR 1.20; 95% CI, 1.05–1.36; *p* = 0.006). At the baseline aortic diameters of 4.5/5.0/5.5 cm, the estimated probabilities of ascending aortic incidences and elective surgical repair within 5 years were 0.4%/1.1%/2.9% and 6.4%/21.2%/51.4%, respectively. The authors concluded that a 5.0 cm cutoff of ascending aortic diameter for prophylactic surgical repair in non-syndromic patients should be considered to lower the risk of aortic dissection and/or rupture. This conclusion decreases the surgery-indicating size of aorta for nearly 1.0 cm since Davies et al. [5] found a rupture or dissection rate of 6.9% and a death rate of 15.6% per year in patients with an aneurysm exceeding 6.0 cm in diameter, and elective operation eliminated the risk of rupture and restored survival to near normal. 

Nevertheless, the above IRAD study also found that mortality was not associated with aortic diameter and did not differ significantly across the 2 to 10 cm subgroups, leading to the difficulty of setting up a single threshold aortic diameter for predicting dissection, rupture, and death risks in all patients [6]. It is notable that undiagnosed patients with inherited TAD and aortitis in the entire population make it even more difficult to set up a precise cutoff point as a surgical indicator because these patients have high risks of dissection and rupture at a small or even normal aortic size. 

#### 1.3.2. Elongation (Longitudinal Enlargement)

In addition to circumferential dilatation, the longitudinal enlargement of proximal aorta has been studied and considered a more reliable risk predictor for aortic dissection due to the fact that aortic diameter enlarges by 16.9% to 31.9%, while the aortic length increases by only 2.7% to 5.4% when dissection occurs [242,243,244]. Despite continuous elongation of the ascending aorta with aging [245,246,247,248], Krüger et al. [246,249,250] found that ascending aortic length was significantly increased in both dissected and pre-dissection aortas when compared with healthy controls, and their studies proposed a score including both ascending aortic diameter (4.5 to 5.4 cm) and length (12 cm) as indicators for preventative surgical repair. 

Wu et al. [244] found that an ascending aortic length of 11 cm could be considered a potential criterion for surgical intervention, which was a better risk predictor than aortic diameter because 70.4% (31/44) of dissected patients had an aortic length over 11 cm, while the aortic diameters were <5.5 cm. Importantly, body height and aortic size are genetically influenced and highly correlated; the same study established a more powerful risk predictor by combining the aortic diameter and length into an aortic height index, sum of aortic diameter plus length then divided by height [244]. The study concludes with a yearly risk estimation chart dividing aortic height index into four categories which are associated with different risks of adverse aortic events, including aortic dissection, rupture, and death. Before this newly established indicator is widely used for making clinical decisions, more patients will be subjected to serial axial and long-term imaging studies to validate this scale in multi-centers. It will also be of interest to test or modify this chart in HTAD patients since many of these patients dissect with smaller aortic size accompanied with skeleton disorder. Interestingly, excessive vertebral artery elongation leads to vessel tortuosity has been quantified by magnetic resonance angiography and a higher vertebral tortuosity index, (actual/straight length-1) × 100, was found associated with younger age at aortic surgery in patients with MFS and Loeys–Dietz syndrome. Moreover, an index of 50 or larger was found a reproducible indicator for early dissection event and mortality [251]. 

Aortic curvature and angulation usually present with ascending aortic elongation due to limited space in the superior mediastinum. In a recent study, Della Corte et al. [252] confirmed aortic elongation in aneurysmal and dissected patients, they also found that the angle between the ascending axis and arch axis was significantly narrower than that in the control aortas and suggested that an ascending-arch angle <130° appears to be a highly sensitive independent predictor of aortic dissection. However, the specificity of the 130°-angle is low, and it requires further study to analyze the pre-dissection data to validate its clinical value. 

## 2. Conclusions

Although substantial progress has been made in improving short- and long-term outcomes of patients with TAD, the mortality remains high mainly due to unpredictable dissection and rupture. Targeting the increasing incidence of AADs needs both better address of known risk factors and further exploration of risks that have not yet been identified. Population-based prospective studies or a combination of current ongoing in-hospital clinical registries and out-of-hospital autopsy data will provide insights into the undetermined area. Importantly, studies with control-improved design based on the current multidisciplinary knowledge will further strengthen the resolution of identifying unknown factors. Finally, there are three high-penetrate and reproducible AAD mouse models that could be used to validate the causative role of uncertain AAD risk factors and to develop potential pharmacologic treatments to prevent AAD. 

## Figures and Tables

**Table 1 genes-13-01814-t001:** General characteristics of type A AAD from Eastern and Western series.

Country/Region or Registry	Sino-RAD	Taiwan	South Korea	Japan	Australia	IRAD	GERAADA	NORCAAD
Year of publication	2014 [104]	2022 [108]	2009 [109]	2022 [107]	2022 [106]	2018 [105]	2014 [110]	2018 [111]
Study time period	2012–2013	2007–2020	1993–2008	2013–2018	2011–2019	1996–2016	2006–2010	2005–2014
Multi-center	Yes	No	No	Yes	No	Yes	Yes	Yes
Number of cases	430	704	256	29,486	98	2952	2137	1131
Age (SD)	50.5 (11.2)	56.6 (13.7)	56 (14)	59.8 (14.2)	64 (14)	61.5 (14.6)	60.5 #	61.6 (12.1)
Male (%)	328 (76.3)	490 (69.6)	139 (54.3)	14,760 (50.1)	63 (64)	1992 (67.5)	1318 (61.7)	757 (66.9)
Hypertension (%)	221 (51.4)	510 (72.4)	134 (52.3)	22,057 (74.8)	73 (75)	2089 (74.4)	1133 (53.0)	587 (52.0)

Data presented as mean (standard deviation) or *n* (%). Sino-RAD, Registry of Aortic Dissection in China; IRAD, International Registry of Acute Aortic Dissection; GERAADA, German Registry for Acute Aortic Dissection Type A; NORCAAD, Nordic Consortium for Acute Type A Aortic Dissection; SD, standard deviation. # Recalculated result based on published data.

## Data Availability

Not applicable.

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
