# Peer review of "Risk Factors for Thoracic Aortic Dissection"

_genes, 2022, doi:10.3390/genes13101814_

Round 1

Reviewer 1 Report

In this review, Zhou and colleagues provide a comprehensive overview of the literature concerning the different risk factors for thoracic aortic aneurysm and dissection (TAD), suggesting novel and interesting elements to take in account in the context of early diagnosis of this condition. The risk factors discussed are classified into i) "General Risk Factors" i.e., associated with unavoidable or random conditions, such as biological sex, genetics, and ethnicity, ii) "Modifiable Risk Factors" i.e., monitorable and therapeutically treatable, such as hypertension, aortitis, and cocaine abuse, and iii) "Other acquired conditions". In addition, the manuscript approaches the important issue regarding aortic dimension and clinical guidelines.

The reading of the manuscript is linear and fluid. The topic is distinctly handled and, despite the large amount of available information, the data presented are consistent and adequate to globally discuss the matter of interest. The structure of the paper is globally logical and analyses the paragraphs from a generical to a more focused point of view, including the molecular mechanism involvements.

This Reviewer has just some concerns that should be solved to improve the manuscript.

Major issues:

-        It could be helpful to list the numerous abbreviations of the manuscript, including those more frequently adopted (e.g., TAD, TAA, AAD, HTAD, IRAD, TEVAR);

-        Re-evaluate chapter "1.1.6. Geography and race": it contains a lot of information, not always clearly expressed. In my opinion, the structure of the paper stands also without this paragraph. Otherwise, please add a table in order to schematically describe this issue (e.g., compare Sino-RAD to IRAD data). In addition, should be payed the attention on the socio-economic discrepancies in the description of the “Black and White differences”;

-        Please, substitute "race" with "ethnicity" at least in the paragraph 1.1.6. title “Geography and race”.

Minor issues:

-        The authors may interestingly state a clear opinion in the end of the paragraph “1.2.5. Fluoroquinolone”, which contains several conflicting data;

-        Please, avoid the contracted form. Replace “it’s” with “it is” (lines 282 and 680);

-        The paragraph “1.4. Protective factor – Diabetes” should be moved after the “1.2.7 Other acquired condition” section. In the opinion of this Reviewer, this could emphasize the positive/protective role of a corroborated cardiovascular risk factor.

Author Response

Response to Reviewer 1 Comments

In this review, Zhou and colleagues provide a comprehensive overview of the literature concerning the different risk factors for thoracic aortic aneurysm and dissection (TAD), suggesting novel and interesting elements to take in account in the context of early diagnosis of this condition. The risk factors discussed are classified into i) "General Risk Factors" i.e., associated with unavoidable or random conditions, such as biological sex, genetics, and ethnicity, ii) "Modifiable Risk Factors" i.e., monitorable and therapeutically treatable, such as hypertension, aortitis, and cocaine abuse, and iii) "Other acquired conditions". In addition, the manuscript approaches the important issue regarding aortic dimension and clinical guidelines.

The reading of the manuscript is linear and fluid. The topic is distinctly handled and, despite the large amount of available information, the data presented are consistent and adequate to globally discuss the matter of interest. The structure of the paper is globally logical and analyses the paragraphs from a generical to a more focused point of view, including the molecular mechanism involvements.

We would like to thank the reviewer for the positive comments on our manuscript.

This Reviewer has just some concerns that should be solved to improve the manuscript.

Major issues:

Point 1: It could be helpful to list the numerous abbreviations of the manuscript, including those more frequently adopted (e.g., TAD, TAA, AAD, HTAD, IRAD, TEVAR);

Response 1: We agree with the reviewer that a list of frequently used abbreviations in the manuscript will help the audience to get the messages more efficiently. We previously followed the Manuscript Preparation instruction of Genes journal - Acronyms/Abbreviations/Initialisms should be defined the first time they appear in each of three sections: the abstract; the main text; the first figure or table. When defined for the first time, the acronym/abbreviation/initialism should be added in parentheses after the written-out form. And we contacted the managing editor and reported this issue during the revising period, without a confirmation yet. We have prepared an abbreviation list and will update the manuscript if it is approved in the final revision.

Point 2: Re-evaluate chapter "1.1.6. Geography and race": it contains a lot of information, not always clearly expressed. In my opinion, the structure of the paper stands also without this paragraph. Otherwise, please add a table in order to schematically describe this issue (e.g., compare Sino-RAD to IRAD data).

Response 2: We regret that the statements were confusing and agree with the reviewer that a summary table will address this issue.   

Point 3: In addition, should be payed the attention on the socio-economic discrepancies in the description of the “Black and White differences”; Please, substitute "race" with "ethnicity" at least in the paragraph 1.1.6. title “Geography and race”.

Response 3: We appreciate the reviewer bringing up this important issue. Based on this feedback and an editorial comment (JAMA. 2021;326(7):621-627. doi:10.1001/jama.2021.13304), we have revised the language in this section to be consistent with originally published records. Hopefully, the revised manuscript will not cause any socio-economic conflict.

Minor issues:

Point 4: The authors may interestingly state a clear opinion in the end of the paragraph “1.2.5. Fluoroquinolone”, which contains several conflicting data;

Response 4: We agree with the reviewer and have removed the observational study that is not consistent with other content in this section.    

Point 5: Please, avoid the contracted form. Replace “it’s” with “it is” (lines 282 and 680);

Response 5: Thank you for pointing out this omission. The manuscript has been checked carefully and the revised manuscript will not contain any similar errors.

Point 6: The paragraph “1.4. Protective factor – Diabetes” should be moved after the “1.2.7 Other acquired condition” section. In the opinion of this Reviewer, this could emphasize the positive/protective role of a corroborated cardiovascular risk factor.

Response 6: We agree with the reviewer and have relocated this paragraph, which will improve the reading experience.

Reviewer 2 Report

This review is  extensive and  comprehensible, and excellently written, reflecting a broad knowledge and insight in the subject. The authors deserve to be congratulated on this achievement.

It is a welcome addition to the TAAD/AAD literature. 

The only comment is that the authors fail to clarify cq discuss that none of the presented  risk factors alone is either a sufficient or necessary cause for AAD/TAAD. And explain to the audience that interaction with a genetic susceptibility (albeit yet mostly unknown) is an important part of the etiology. 

Therefor the authors are recommended to add some lines to the manuscript so that the audience does understand the etiologic complexity, that AAD/TAAD like many other (adult onset) disorders can only be explained by complex interactions between genetic and other risk factors.   

Author Response

Response to Reviewer 2 Comments

This review is extensive and comprehensible, and excellently written, reflecting a broad knowledge and insight in the subject. The authors deserve to be congratulated on this achievement. It is a welcome addition to the TAAD/AAD literature. 

We thank the reviewer for the positive comments on our revised manuscript.

Point 1: The only comment is that the authors fail to clarify cq discuss that none of the presented risk factors alone is either a sufficient or necessary cause for AAD/TAAD. And explain to the audience that interaction with a genetic susceptibility (albeit yet mostly unknown) is an important part of the etiology. Therefor the authors are recommended to add some lines to the manuscript so that the audience does understand the etiologic complexity, that AAD/TAAD like many other (adult onset) disorders can only be explained by complex interactions between genetic and other risk factors.   

Response 1: We agree with the comments by this reviewer. The fundamental role of genetic variation is emphasized and other presented risk factors that will work as a second hit, but not alone, in the occurrence of thoracic aortic dissection, as stated in the introduction as follows:

Additionally, up to 80% of individuals presenting with AADs do not have a known family history of aortic disease or a pathogenic variant in an established TAD gene, and very little is understood as to why these AADs occur. These individuals are hypothesized to harbor one or more genetic variants that predispose them to AADs, and in combination with environmental insults or a second genetic hit, trigger AADs. This review will focus on the risk factors that have not been proved to cause AADs solely, and how environmental and lifestyle risk factors may combine with genetic variants to trigger AADs.

Round 2

Reviewer 1 Report

This Reviewer accepts the Author's corrections and considers this manuscript appropriate for the publication.